# Physiological Profile Assessment and Self-Measurement of Healthy Students through Remote Protocol during COVID-19 Lockdown

**DOI:** 10.3390/jfmk9030170

**Published:** 2024-09-19

**Authors:** Tommaso Di Libero, Lavinia Falese, Annalisa D’Ermo, Beatrice Tosti, Stefano Corrado, Alice Iannaccone, Pierluigi Diotaiuti, Angelo Rodio

**Affiliations:** Department of Human, Social and Health Sciences, University of Cassino and Southern Lazio, Campus Folcara, Via S. Angelo, 03043 Cassino, FR, Italy; tommaso.dilibero@unicas.it (T.D.L.); l.falese@unicas.it (L.F.); beatrice.tosti@unicas.it (B.T.); stefano.corrado@unicas.it (S.C.); alice.iannaccone@unicas.it (A.I.); p.diotaiuti@unicas.it (P.D.); a.rodio@unicas.it (A.R.)

**Keywords:** COVID-19, physical activity, fitness level, functional assessment, remote testing, field test

## Abstract

**Background**: The COVID-19 pandemic has led to reduced physical activity and increased sedentary behaviors, negatively impacting mental and physical health. Engaging in physical activity at home during quarantine became essential to counteracting these adverse effects. To develop appropriate physical activity programs, assessing individuals’ fitness levels and the impact of inactivity is crucial. This study aims to compare motor abilities—including flexibility, balance, reaction time, cardiovascular endurance, and lower and upper limb strength—assessed both in-person and remotely, to determine the accuracy and repeatability of self-administered tests. **Methods**: A total of 35 young subjects (age 24.2 ± 1.97 years, BMI 22.4 ± 2.61 kg/m^2^) participated in this study. Each participant underwent a battery of motor ability tests designed to assess various fitness components. The tests were administered twice for each subject: once in a laboratory setting and once remotely at home. The sequence of tests was randomly assigned to ensure unbiased results. Both the in-person and remote assessments were used to evaluate the accuracy and reliability of self-administered tests. **Results**: The comparison of test results between the laboratory and remote settings revealed percentage differences ranging from 5% to 10%. This variation is considered an acceptable margin of error, suggesting that the tests conducted remotely were relatively accurate when compared to those performed in a controlled laboratory environment. **Conclusions**: The findings indicate that remote fitness testing is a promising method for evaluating motor abilities. With an acceptable margin of error, remote assessments can be effectively used to personalize training programs based on individuals’ physiological characteristics. This approach may be particularly beneficial during times of limited access to fitness facilities, such as during quarantine, or for individuals seeking more flexible fitness evaluation methods.

## 1. Introduction

Physical fitness level is an ability to carry out daily tasks with vigor and alertness without undue fatigue [1], and it is an important predictive factor of the wellness and health of an individual [2]. Physical fitness involves different health-related anthropometric and physiological parameters such as body composition, muscle power, cardiorespiratory stress, and osteoarticular capacities (reactivity, flexibility, and balance) [3]. According to the World Health Organization (WHO) [4], regular physical activity (PA) and healthy lifestyle behaviors [5] represent a non-pharmacological approach, a useful tool for prevention and therapy to promote health with different benefits in respiratory, circulatory, and immune function systems, as well as a reduction in cardiovascular risks and other non-communicable diseases [6,7,8]. PA is crucial for maintaining good health and well-being [9,10]. Importantly, PA significantly improves and maintains physical fitness levels by developing motor abilities, endurance, strength, and flexibility [11]. It significantly reduces the risk of cardiovascular disease by improving blood circulation and heart function [12]. Additionally, exercise promotes efficient metabolism, which helps regulate body weight and improve blood sugar control [13]. To date, in order to obtain the positive effects on wellness and health induced by PA [14], the guidance for the adult population, in particular, is that they should engage in a total PA of 150–300 min/week of moderate exercise or at least 75–100 min/week of vigorous exercise, and/or combination of these [15], which should correspond to about 1500 Kcal/week.

Due to the COVID-19 spread, which occurred in February 2019, numerous problems affected the whole world, which led to restrictive actions being implemented by governments for the entire population. This resulted in negative consequences for PA due to the social distancing and self-isolation at home that was imposed to prevent the spread of the virus [16]. The main consequences of a prolonged homestay affected socialized indoor PA activities (e.g., dance, yoga, and gym) and other examples of PA such as swimming, football, and other team sports [17]. Furthermore, the closing of public places used for social activities, i.e., parks and community centers, led to a subsequent reduction in regular PA, including recreational activity, often resulting in major psychological distress, including symptoms of mental illness such as depression, anxiety, insomnia, and psychophysical high stress levels [18,19]. It can also lead to a compromised immune system, a heightened risk of cardiovascular disease, and a range of other adverse health effects [20]. This is due to hypothalamic–pituitary–adrenal axis activation, which increases cortisol levels, leading to chronic inflammation, insulin resistance, and obesity [21]. Several studies have identified factors associated with worsening clinical outcomes among patients with COVID-19, including pre-existing comorbidities (e.g., pulmonary disease and heart disease) [22], negative lifestyle (e.g., smoking and diet), and demographic characteristics (e.g., male sex and older age) [23,24]. There has also been the identification of protective factors between cardiorespiratory fitness and health outcomes, with the inverse relationship of maximal exercise capacity and a lower risk of severe illness due to COVID-19 being found [25]. Social restriction due to the COVID-19 lockdown has the amplified negative effects induced by reduced PA, and it is also directly related to an imbalance between energy intake and expenditure, implying fat accumulation [26].

To contrast these problematic issues linked to COVID-19, including physical and psychological stress, many operators have started to carry out remote training and activities by means of web platforms and social tools to perform PA [27]. Although the new strategies of remote training during the COVID-19 lockdown resulted in a very good solution, there is a need to find similarly new remote modalities for functional assessment to make training monitorable and individualized and to meet the time recommendations necessary to achieve the positive effects of PA [28]. One of the primary challenges in remote assessment is to find an effective system to optimally tailor physical-activity-based interventions. The ability to obtain detailed information about an individual’s physiological profile through body composition assessments of coordinative and conditional abilities is the basis of creating personalized and effective training strategies [29,30]. One of the challenges of remote training is the difficulty of monitoring during performing exercises, as well as determining the appropriate loads, volumes, and training schedules. This can only be conducted effectively when the individual’s anthropometric and physiological profile has been assessed. Previous studies, through simple tests assessing both motor and cognitive parameters, have reported the utilization of sensorized devices, questionnaires, or neurofeedback systems, allowing for the effective assessment of subject abilities remotely [31,32,33]. Therefore, it is necessary to propose evaluation tests and investigate whether these assessments’ reliability remains valid when conducted remotely.

The primary objective of this study was to perform simple remote assessments to evaluate body composition and coordinative and functional abilities using easily repeatable field tests. To assess coordination, the following tests were applied: the V Sit and Reach (VS&R) for flexibility, the Stork Balance Test (SBT) for balance, and the Ruler Drop Test (RDT) for reaction time. Functional abilities were evaluated using Ruffier’s test (RT) for cardiovascular endurance, along with the Squat Jump (SJ) and Push Up Test (PUp) for lower and upper limb strength, respectively. These tests were suitable for home use as they did not require specialized equipment. This methodology allowed us to establish the consistency of the results in two different scenarios, specifically in laboratories (L1 and L2) and in remote settings (R1 and R2). Comparing these assessment methods is particularly important in situations like the COVID-19 lockdown when access to physical testing facilities is limited. This can be also useful for remote assessments in sports or motor reconditioning as remote testing may be a valid method for tailoring training programs to individual physiological characteristics.

## 2. Materials and Methods

### 2.1. Participants

In this work, a population of 35 bachelor’s and master’s degree exercise science students at the University of Cassino and Southern Lazio, participating in an internship activity, were enrolled. Specifically, the sample was composed of 17 females and 18 males, aged between 21 and 30 years. All participants were informed about how the protocol would be carried out and provided their consent before participating in the study. Moreover, informed consent and authorization about benefits and risks were obtained in accordance with the Declaration of Helsinki for Human Research of 1964. This work was approved by the Institutional Review Board of the University of Cassino and Southern Lazio (no. 24777.2022.12.12). On the familiarization day, all subjects were given anthropometric measurements and body composition assessment, the data of which are shown in Table 1.

### 2.2. Protocol

For body composition, the girth method was used [34], consisting of an equation to determine body fat percentage [35], which requires inputting three distinct constants. These constants correspond to the measurements of body circumferences from three specific landmarks, as shown in Figure 1. All participants were asked not to perform any training sessions during the 10-day study period, during which measurements were taken, and to wear sports clothes. According to the following scheme, test scenario sequences were defined randomly using the same software used for the statistical analysis (Statistical Package for Social Science: SPSS), ensuring that each subject performed the test session twice at home and twice in the laboratory.

First day: A laboratory familiarization session was held to ensure adequate training for participants and to be sure that test execution was clearly defined for all participants. Overall, the system measures HR safely by placing the index and middle fingers on the neck, to the side of the windpipe, without too much pressure to avoid fainting due to the carotid body reflex.Second day: Previously, the supervisors recorded tutorial videos for the Home Familiarization, in which they explained the correct execution of battery tests and heart rate (HR) measurements at three different times (pre-test, immediate post-test and one-minute post-test).Third day: No activity was held during the day, as well as days 5, 7 and 9.Fourth day (as well as days 4, 6, 8 and 10): The battery test was performed, including tests to measure conditional and coordinative abilities. All participants have been engaged in remote or in laboratory execution of submaximal tests to assess physiological parameters after a 10-min warm-up, consisting of:

VS&R [36,37], for hamstring and low back flexibility. The V Sit and Reach involves sitting on the floor with legs extended in a “V” shape, with feet 20–30 cm apart. A measuring tape should be placed between the legs, with the zero point aligned with the heels. While keeping the knees straight, the individual reaches forward with both hands toward their toes, measuring flexibility;

SBT [38] to assess static balance. The individual stands on one leg with the other leg bent, and the duration for which they can maintain this position is recorded;

SBT was performed barefoot, with eyes open and the hands on the hips. Participants were instructed to stand on one foot, positioning the free foot on the knee of the standing leg. Participants have to lift their heels and stay on the forefoot at the go signal. The test ends when the heel of the supporting leg touches the ground or the foot moves away from the knee and the participant loses their balance [38];

RDT [39] measures the speed of response. A ruler is dropped between the individual’s open fingers, and they must catch it as quickly as possible. The distance the ruler falls before being caught provides a measure of reaction time. When performed at home, a cohabitant of the subject was asked to play the role of the operator; if no one was available, the test could not be performed;

RT [40] evaluates cardiovascular endurance. The individual performs 30 squats in 45 s, with heart rate (HR) measured in three different moments: pre-test, immediately after, and one minute after. Pre-test heart rate was measured in a standing position after a five-minute rest period in a supine position, as well as immediately after and one minute after [41]. The variations in HR indicate cardiovascular recovery capacity and endurance. The RT is a straightforward and valid method for assessing cardiovascular fitness. Compared to others, this submaximal test is easy to reproduce and, aside from a timer, does not require any special equipment unlikely to be found in people’s homes, as a step or box of a specific height [41];

SJ [42] to evaluate the explosive power of the legs is assessed by having the individual perform a jump from a squatting position and measuring the height of the jump. This test determines the explosive capability of the legs. When performed at home, each participant recorded the jump using a smartphone. Afterward, the operators evaluated the jump height using the mobile app My Jump 2 (My Jump 2, Carlos Balsalobre-Fernández, Alcalá de Henares, Madrid, Spain) [43].

PUp [44] measures upper body muscular strength and endurance. The number of correctly performed push-ups within a specified time frame or until exhaustion is counted. This test evaluates both muscle strength and endurance. When performed at home, a plastic water bottle was positioned underneath the participant’s chest to ensure the subject performed correctly (supervisors checked that the participant’s chest touched the plastic bottle and then returned to a fully extended elbow position). The same activity was repeated during days 4, 6, 8 and 10, as shown in Figure 2.

VS&R, SB, RDT and SJ were performed three times, and the best results were taken into analysis, while RT and PU were performed once. When performed in the laboratory, the protocol was conducted in a traditional set-up: the operator assisted participants and administered the battery test. Otherwise, when performed at home, the session was conducted remotely on the Google Meet Platform (Google Meet, Google LLC, Mountain View, CA, USA). Participants underwent self-tests in their own suitable homeroom, equipped with free space and an active internet connection, under the supervision of qualified sport science experts, to avoid possible injuries and failure test due to incorrect posture or movement during evaluation sessions [45]. The testing session takes 15 to 20 min for each participant in both settings. The experiment began on 9 January 2023 and lasted 4 months, following approval by the Ethics Committee on 12 December 2022. By this time, the Italian government had begun to lift the state of emergency, allowing participants to attend data collection sessions in person.

### 2.3. Statistical Analisis

Descriptive statistics was performed for anthropometric data, mean and standard deviation values (or median and interquartile range values) are computed to characterize all students. The Shapiro–Wilk/Kolmogorov–Smirnov normality test was performed to assess the normality distribution of all variables in both males and females. According to the result of the normality test, the two-tailed Pearson or the Spearman correlation analysis was conducted. If measured variables were normally distributed on the different test results, the one-way ANOVA and, if it was significant (p≤0.05), a post hoc test using Bonferroni correction (p≤0.0175) was performed to investigate in which pairwise comparisons of the groups by session the significant differences lie. If the normality test was not satisfied, the correspondent non-parametric analysis, including the Kruskal–Wallis test and Mann–Whitney U-test for all pairwise comparisons, was performed. In addition, we conducted an analysis of the statistical power for our study (Spearman’s and Kendall’s nonparametric tests), and it has been confirmed that a sample size of 35 participants meets the criteria for adequate statistical power at 0.8. This ensures that the sample size is sufficient to detect a significant effect, as recommended by statistics. The magnitude of the correlations was classified according to the following criteria: negligible (<0.1), small (0.1 to 0.29), moderate (0.3 to 0.49), large (0.5 to 0.69), very large (0.7 to 0.89) and near-perfect (≥0.9 to 1) [46,47]. For the statistical analysis, the IBM^®^ SPSS^®^ Statistics 25.0 (SPSS, IBM, Armonk, NY, USA) software was used, assuming the significance level p≤0.05.

## 3. Results

Anthropometric data measured in the first familiarization day are shown in Table 1. BMI ranged between 18.5 and 24.9, resulting in normal weight values (22.4 ± 2.61) for FM% (19.6 ± 7.19). Regarding the coordinative and conditional abilities tests, results are reported in Table 2. The sample showed below-average VS&R and SBT, an above-average value in RDT, low values in RT, excellent performance in SJ, and good results in PUp, with body composition assessment conducted in the laboratory by circumferential measurements. When comparing tests performed at home and remotely, all the results were not statistically significant (*p* > 0.05), as shown in Table 3. Furthermore, the percentage difference between the means for each test was calculated in order to estimate the error percentage of each measurement and evaluate the reliability of remote testing. Since the error percentage is, in all cases, between −10% and 10%, it is possible to say that these measurements are sufficiently reliable [48]. In addition, as seen in Table 3, all percentages are less than −5% and 5%, meaning that the data are included in the limits of agreement: 95% is the likely range of change in measurements between two trials [49].

## 4. Discussion

The main purpose of this study was to compare battery tests for body composition and motor abilities assessment in two different scenarios: in a laboratory setting in which operators measured the variables, and at home with remote assistance performed by operators to assess whether they had the same evaluative value and reliability when administered. This comparison is particularly relevant in contexts such as COVID-19 quarantine, where the need for remote solutions has become essential, the same as for education and healthcare, especially in cases where patients are unable to travel to outpatient clinics or service delivery facilities. The results obtained from the descriptive analyses showed that all subjects were within a normal weight range.

Regarding coordinative and conditional abilities, the results were comparable between the two different scenarios. Percentage differences were found, with variations between −10% and 10%, highlighting repeatability and accuracy. These differences suggest that tests depending on the time factor as SBT, are sensitive to small differences. Indeed, a 1 s difference in test results is enough to double the percentage difference. Therefore, we can hypothesize that the minor differences are on the SBTL as the time spent in the requested position was short in comparison with the right side for all the participants. This could be due to the fact that all participants are right-handed, so there was more variability on the right side, evolving into a major percentage difference. A specular situation is depicted in RDTL. In this scenario, higher variability is due to the non-dominant hand’s minor reactivity compared to the dominant hand. The results showed greater efficiency in achieving better performance with the dominant hand. This may be due to the neuromuscular system’s better motor task ability in the dominant hand than in the non-dominant hand [50]. Despite the clear percentage difference, the *p*-value results suggest a similar data distribution in both cases, confirming the remote test reliability.

The RDT has a main limitation as it requires an operator to be present, which is not always possible during lockdown or quarantine. An alternative could be the tapping test, which, in its simplicity, can provide accurate information about the neuromuscular system [29]. Research has shown that the tapping test can be performed independently by the individual without the need for complex equipment or continuous supervision, making it particularly useful in isolation settings. Additionally, the tapping test is sensitive to changes in neuromuscular function, allowing reliable and repeatable assessment of the motor system’s status, even under conditions of limited access to healthcare facilities. These findings are crucial, as confirm the validity and reliability of the tests even in remote mode.

### 4.1. Strengths of the Study

Our results may have significant implications, not only in the sports context but also in a pathological one. For example, several studies that considered a pool of cancer survivor patients conducted during the pandemic have reported similar results [51,52,53,54,55,56]. The administrated tests were those utilized for the elderly in particular, such as the Time Up and Go, the Six-Minute Walking Test, and the facilitated test to evaluate static balance. Maintaining a physical exercise routine is essential to preserve a fitness status for health and wellness, especially in the elderly. Situations such as the pandemic may have adverse effects on elderly individuals who are confined to their homes, making adapted PA practices extremely important [57,58]. Our proposed tests, such as the Ruffier, STB and S&R, do not require specific settings or conditions, unlike tests for the elderly or individuals with medical conditions, as seen in previous cited works. To tailor a personalized and safe PA program, it is crucial to carry out an evaluation of fitness level to identify the intensity and frequency of workload. This can be achieved by following the methodology outlined in this work or, in addition, by utilizing advanced technology when addressing pathological conditions that demand particular attention to specific factors [59]. Thanks to technological advancements, it is now possible to integrate devices that can be remotely used to conduct more affordable assessments [60,61]. Furthermore, a remote method should be considered to ensure cognitive function training. Neurofeedback devices in distance learning programs are highly beneficial, as they allow for precise and personalized assessment of brain activities. This, in turn, improves the effectiveness of therapeutic interventions without the need for physical presence [62,63]. These devices can generate an extensive database, providing physicians with motor and cognitive crucial information to monitor patients [64,65]. These methodologies can be valuable for adjusting treatments in cases where patients are unable to access specialized centers for their care. This work provides a solid foundation for the adoption of remote assessment practices in various settings, helping to maintain high standards of monitoring and intervention despite the limitations imposed by extraordinary circumstances such as a pandemic.

### 4.2. Limitations

While the results obtained are promising, it is important to acknowledge the limitations of the present study. Firstly, the relatively small sample size may restrict the generalizability of the findings. Conducting future studies with a larger sample size will be essential to ensure the robustness of the conclusions. Additionally, since the current sample predominantly comprises young students, it is necessary to consider how this may impact their response to the assessment protocol. Therefore, including a broader variety of age groups in future participants will be crucial to determine whether the observed results remain consistent across different age demographics. This expansion of the sample will provide a more comprehensive understanding of whether the proposed protocol can be effectively applied in a range of settings beyond those included in the present study.

## 5. Conclusions

The pandemic of COVID-19 has highlighted the need for innovative methods for remote assessments. This work demonstrates significant advances in accurate remote assessments, enabling professionals to monitor motor abilities safely and accurately. The replicability of remote testing ensures the continuity of essential assessments even under restrictive conditions. Future implications include the exploration of additional reliable remote testing and the integration of artificial intelligence. The results suggest that institutions can confidently invest in remote testing technologies, optimizing resources and providing greater flexibility in test management.

## Figures and Tables

**Figure 1 jfmk-09-00170-f001:**
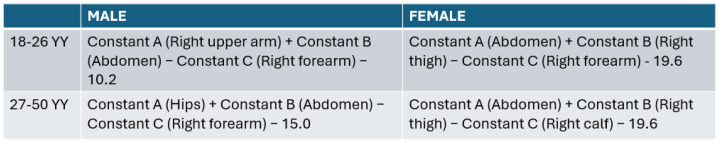
Constants (A, B, C) related to body landmarks and equations to calculate body fat percentage (FM%).

**Figure 2 jfmk-09-00170-f002:**
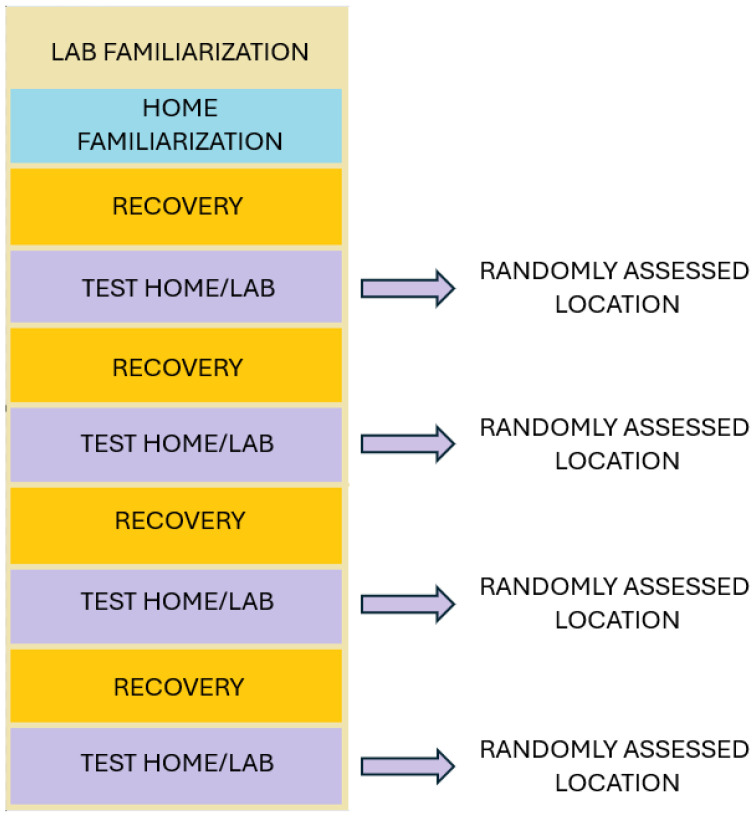
Protocol setting.

**Table 1 jfmk-09-00170-t001:** Participants anthropometric data. BMI = Body Mass Index; FM = Fat Mass.

				Shapiro–Wilk
	Gender	Mean	SD	*p*
Age (y)	F	24.4	1.62	0.013
	M	24.1	2.29	0.260
Weight (kg)	F	58.8	10.49	0.671
	M	73.2	6.81	0.007
Height (cm)	F	165.5	6.96	0.333
	M	177.1	8.57	0.651
BMI (kg/m^2^)	F	21.4	3.04	0.514
	M	23.4	1.73	0.504
FM (%)	F	23.6	6.5	0.421
	M	15.9	5.74	0.238

**Table 2 jfmk-09-00170-t002:** Coordinative and conditional abilities results with mean and relative standard deviation. The data provided refer to the tests performed in the laboratory (L1 and L2) and in remote (R1 and R2). L = Laboratory; R = Remote; VS&R = V Sit and Reach; SBTL = Stork Balance Test Left; SBTR = Stork Balance Test Right; RT = Ruffier’s Test; SJ = Squat Jump Test; PUp = Push Up Test.

	L1	R1	L2	R2
VS&R (cm)	37.6 ± 16.83	38.5 ± 16.65	38.1 ± 16.65	39.0 ± 16.56
SBTR (s)	4.2 ± 2.6	4.0 ± 2.74	4.8 ± 3.05	4.7 ± 2.90
SBTL (s)	4.1 ± 2.91	4.2 ± 2.98	4.1 ± 3.09	4.4 ± 3.01
RDTR (cm)	14.6 ± 10.9	13.8 ± 10.6	14.5 ± 10.6	14.3 ± 10.62
RDTL (cm)	15.9 ± 9.66	14.1 ± 8.98	14.8 ± 10.67	15.0 ± 9.79
RT (ua)	7.7 ± 3.59	7.1 ± 2.98	7.0 ± 2.67	6.8 ± 2.55
SJ (cm)	29.9 ± 8.17	30.5 ± 7.75	31.6 ± 7.28	31.8 ± 6.96
PUp (rep)	24.6 ± 11.2	25.2 ± 11.3	25.3 ± 10.8	25.6 ± 11.5

**Table 3 jfmk-09-00170-t003:** Error percentage between the mean of measurements for each test; in brackets the *p*-value, set to >0.005, from the comparison between all modalities measurements (L = Laboratory; R = Remote; VS&R = V Sit and Reach; SBTL = Stork Balance Test Left; SBTR = Stork Balance Test Right; RT = Ruffier’s Test; SJ = Squat Jump Test; PUp = Push Up Test).

	L1 vs. R1	L1 vs. R2	L1 vs. L2	L2 vs. R1	L2 vs. R2	R1 vs. R2
			Diff% (*p*)			
VS&R (cm)	2.39 (0.263)	3.72 (0.060)	1.33 (0.768)	1.05 (0.699)	2.36 (0.244)	1.30 (0.845)
SBTR (s)	−4.76 (0.937)	11.90 (0.367)	14.29 (0.375)	−16.67 (0.062)	−2.08 (0.997)	17.50 (0.080)
SBTL (s)	2.44 (0.974)	7.32 (0.947)	0.00 (1.000)	2.44 (0.972)	7.32 (0.682)	4.76 (0.985)
RDTR (cm)	−5.48 (0.642)	−2.05 (0.914)	−0.68 (0.999)	−4.83 (0.721)	−1.38 (0.947)	3.62 (0.878)
RDTL (cm)	−11.32 (0.104)	−10.69 (0.084)	−6.92 (0.083)	−4.73 (0.357)	−4.05 (0.301)	0.71 (0.843)
RT (ua)	2.01 (0.472)	6.35 (0.473)	5.69 (0.530)	−3.48 (0.989)	0.63 (0.984)	4.26 (0.880)
SJ (cm)	−7.79 (0.818)	−11.69 (0.179)	−9.09 (0.279)	1.43 (0.517)	−2.86 (0.956)	−4.23 (0.390)
PUp (rep)	2.44 (0.518)	10.98 (0.987)	2.85 (0.973)	−0.40 (0.214)	7.91 (0.904)	8.33 (0.656)

## Data Availability

Data are contained within the article.

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
