# Peer review of "Physiological Profile Assessment and Self-Measurement of Healthy Students through Remote Protocol during COVID-19 Lockdown"

_jfmk, 2024, doi:10.3390/jfmk9030170_

Round 1
Reviewer 1 Report
Comments and Suggestions for Authors
Review for: "Physiological Profile Assessment And Self-Measurement Of Healthy Students Through Remote Protocol During Covid-19 Lock Down"
The COVID-19 pandemic has led to more sedentary behaviors due to restrictive measures, negatively affecting mental and physical health. This study aimed to compare the results of motor abilities tests conducted in-person and remotely to determine if they maintain their evaluative value.
Below I send you my comments that can help improve the quality of the work:
1. In the title of the work, it is necessary to correct "Healty" to "Healthy".
2. It is necessary to rewrite the abstract. More than half of the abstract is theory, and the results have a negligible part. 2-3 sentences of the introduction, then 1 sentence of the goal, 1-2 sentences of the methodology and the rest should be the results and conclusion of the work, as well as recommendations for further activities
3. Psychological stress was mentioned in the introduction of the paper, but not how it affects physical health. It is necessary to significantly supplement this part.
4. It is not clearly explained how physical activity affects health. This should be stated concisely and precisely.
5. line 84-86 "Comparing these assessment methods is particularly important in situations like Covid-19 lockdown or can be useful for assessment in sport, clinical rehabilitation or research." This sentence is in the introduction of the paper, but it is not explained. Why the sample is students who are healthy, and the results favor patients in rehabilitation or other groups. The introduction to the work is not good, it needs to be completely rewritten and concisely addressed all the factors related to the goal of the work.
6. In the methodology, the time of conducting the study is not specified anywhere.
7. line 122-155. None of the tests applied in the paper were mentioned in the introduction or explained in more detail in the methodology. All this can be done in a couple of sentences without overloading the text. Please make the appropriate additions.
8. line 167-168 "An observational study was conducted on the entire sample of 35 students to analyze body composition." This sentence should not be in the statistical analysis section.
9. line 166-167 "For the statistical analysis the IBM® SPSS® Statistics 25.0 (SPSS, IBM, USA) software was used, assuming the significance level p ≤ 0.05." should be at the end of the paragraph.
10. I advise that tables 2, 3, 4 be combined into one for the sake of transparency of the data.
11. I think you can do some additional statistical tests to supplement the scientific weight of your work. There are very few results displayed this way.
12. Strengths and limitations should be a separate paragraph in the discussion.
13. The conclusions should be rewritten. Highlight the most important parts of the work and give recommendations for further activities. Written like this, the format of the conclusion is not good.
In general, throughout the entire paper, the authors did not emphasize the results of their work, which are lost in a lot of text and bad sentence constructions.
I think that with major revisions, this paper can gain more scientific weight and be of better quality.
Comments on the Quality of English LanguageModerate editing of English language required.
Author Response
COMMENT 1: In the title of the work, it is necessary to correct "Healty" to "Healthy".
RESPONSE 1: Thank you for noticing the typo. We have corrected it.
COMMENT 2: It is necessary to rewrite the abstract. More than half of the abstract is theory, and the results have a negligible part. 2-3 sentences of the introduction, then 1 sentence of the goal, 1-2 sentences of the methodology and the rest should be the results and conclusion of the work, as well as recommendations for further activities
RESPONSE 2: Thank you for your valuable comments. We have edited the abstract following your suggestions.
COMMENT 3: Psychological stress was mentioned in the introduction of the paper, but not how it affects physical health. It is necessary to significantly supplement this part.
RESPONSE 3: Thank you for your insightful feedback. We appreciate your suggestion regarding the clarification of how physical activity affects health. As recommended, we have expanded the part explaining the effects due to stress.
COMMENT 4: It is not clearly explained how physical activity affects health. This should be stated concisely and precisely.
RESPONSE 4: Thank you for your valuable feedback. We appreciate your suggestion regarding the clarification of how physical activity impacts health. As recommended, we have expanded the explanation of the effects of stress.
COMMENT 5: line 84-86 "Comparing these assessment methods is particularly important in situations like Covid-19 lockdown or can be useful for assessment in sport, clinical rehabilitation or research." This sentence is in the introduction of the paper, but it is not explained. Why the sample is students who are healthy, and the results favor patients in rehabilitation or other groups. The introduction to the work is not good, it needs to be completely rewritten and concisely addressed all the factors related to the goal of the work.
RESPONSE 5: Thank you for your feedback. It's true that the sentence was unclear. The study aimed to assess whether field tests, which are typically conducted in person, could also be conducted remotely without an operator present. These field tests vary depending on the individual's age or physical condition. In our investigation, we focused on tests designed for healthy young adults, for which we recruited healthy students. We revised the highlighted sentence to improve its clarity.
COMMENT 6: In the methodology, the time of conducting the study is not specified anywhere.
RESPONSE 6: Thank you for noticing the lack of information. We added information about the time spent to administer the test and to carry out the experimentation right at the end of the section “protocol” (line 154).
COMMENT 7: line 122-155. None of the tests applied in the paper were mentioned in the introduction or explained in more detail in the methodology. All this can be done in a couple of sentences without overloading the text. Please make the appropriate additions.
RESPONSE 7: Thank you for noticing the missing test description. Now, You can find the description of VS&R highlighted in bold from line 120 to line 124. As for the other tests described immediately following the VS&R. Specifically the SBT from lines 125 to 131, the RDT from lines 132 to 136, the Ruffier test from lines 137 to 145, the SJ from lines 146 to 150, and the Pup from lines 151 to 156. In addition, we followed your advice and mentioned the tests in the introduction as well, from line 71 to 78.
COMMENT 8: line 167-168 "An observational study was conducted on the entire sample of 35 students to analyze body composition." This sentence should not be in the statistical analysis section.
RESPONSE 8: Thank you for the comment, you are right, it was a typo that we had not noticed. We have now removed the sentence from the section.
COMMENT 9: line 166-167 "For the statistical analysis the IBM® SPSS® Statistics 25.0 (SPSS, IBM, USA) software was used, assuming the significance level p ≤ 0.05." should be at the end of the paragraph.
RESPONSE 9: Thank you for the suggestion. We have moved the sentence as indicated.
COMMENT 10: I advise that tables 2, 3, 4 be combined into one for the sake of transparency of the data.
RESPONSE 10: Thank you for your excellent advice. Whilst it is possible to combine Tables 3 and 4 as we have done, it is unfortunately not possible to include Table 2 as it relates to a different parameter.
COMMENT 11: I think you can do some additional statistical tests to supplement the scientific weight of your work. There are very few results displayed this way.
RESPONSE 11: Thanks for your suggestion. Spearman's and Kendall's nonparametric analysis have been added, and we have included additional detail in the methods of our statistical analysis.
COMMENT 12: Strengths and limitations should be a separate paragraph in the discussion.
RESPONSE 12: Thank you once again for yet another helpful suggestion. We reorganised the discussion as requested
COMMENTI 13: The conclusions should be rewritten. Highlight the most important parts of the work and give recommendations for further activities. Written like this, the format of the conclusion is not good.
RESPONSE 13: Thank you for the suggestion. We have rewritten and summarised the conclusions according to your suggestions.

Reviewer 2 Report
Comments and Suggestions for Authors
Dear authors,
Thank you for the opportunity to evaluate your paper titled: Physiological Profile Assessment And Self-Measurement of Healty Students Through Remote Protocol during COVID-19 Lock Down
Follow my reviews:
Introduction
Please, let clear the level of novelty of your study. What is the new content that it brings for the field?
Participants
Please, report the statistical power for included sample.
Table 1
Table 1 could be more informative if you can present comparisons between male and female gender. Please, adjust it.
Discussion
The discussion needs to be divided in paragraphs to more readable. Further, some sentences must be rewrote, like: For 236 example, several studies that considered a pool of cancer survivors patients conducted 237 during the pandemic have reported similar results [42,43] - You report that "several studies have reporting similar results; however you cite only two studies. Please, reorganize coherence and cohesion along your discussion.
Add limitations and strengths in the end of the discussion. Finally, add research perspectives.
Conclusion
Summarize the conclusion. It is too long.
Comments on the Quality of English Language
English needs an extensive review to improve objectivity and text clarity.
Author Response
Introduction
COMMENT 1: Please, let clear the level of novelty of your study. What is the new content that it brings for the field?
RESPONSE 1: Thank you for your comment. We examined the feasibility and accuracy of remotely administered physical tests compared to those conducted in the laboratory. This comparison allowed us to demonstrate that self-administered tests can achieve an acceptable margin of error (5% to 10%) compared to in-person tests. This evidence is particularly significant in the context of the COVID-19 pandemic, where access to physical testing facilities has been limited. In addition, the results suggest that remote testing may be a valid method for tailoring training programs to individual physiological characteristics. Our research, therefore, not only confirms the reliability of remote testing, but also highlights its potential in the area of personalizing fitness interventions during restrictive situations.
Participants
COMMENT 2: Please, report the statistical power for the included sample.
RESPONSE 2: Thanks for your comment. Spearman's and Kendall's nonparametric analysis have been added, and we have included additional detail in the methods of our statistical analysis.
Table 1
COMMENT 3: Table 1 could be more informative if you can present comparisons between male and female gender. Please, adjust it.
RESPONSE 3: Thank you for your comment, we reworked Table 1 as requested.
Discussion
COMMENT 4: The discussion needs to be divided in paragraphs to more readable. Further, some sentences must be rewrote, like: For 236 example, several studies that considered a pool of cancer survivors patients conducted 237 during the pandemic have reported similar results [42,43] - You report that "several studies have reporting similar results; however you cite only two studies. Please, reorganize coherence and cohesion along your discussion.
Add limitations and strengths in the end of the discussion. Finally, add research perspectives.
RESPONSE 4: Thank you once again for yet another helpful suggestion. We supplemented the text section with additional references to that topic to give more coherence to what has been said and we reorganised the discussion as requested.
Conclusion
COMMENT 5: Summarize the conclusion. It is too long.
RESPONSE 5: Thank you for the suggestion. We have rewritten and summarised the conclusions according to your suggestions.

Reviewer 3 Report
Comments and Suggestions for Authors
Thank you for opportunity for review of this study.
In the introduction, paragraphs should be divided around the central sentence for readability.
After defining physical fitness, the definition of physical activity is mentioned, so content on the relationship between the two should be added.
Before the fact that there was a decrease in PA during COVID-19, the basis for problems that may arise due to the decrease in PA should be presented.
Present a specific research hypothesis or research question.
Can you provide information on how well the participants learned the measurement method through training?
Since the degree of familiarity with the measurement method through training can affect the research results, it is thought that information on this should be provided.
More detailed descriptions should be given on how randomization was done.
As with the introduction, the results and discussion should be divided into paragraphs centered around the main results.
Can you see the difference in results by gender?
Can you see the difference by bachelor and master?
The above information is thought to be helpful in suggesting specific measures.
A note should be provided on the abbreviations used in the table.
The detailed discussion of results should be omitted from the discussion.
There is no section on limitations of the study and suggestions for investing in platforms and technologies for remote test administration
Author Response
COMMENT 1: In the introduction, paragraphs should be divided around the central sentence for readability.
RESPONSE 1: Thank you for the advice, we have made the changes you requested. Effectively the text in one paragraph was taking on an unreadable appearance.
COMMENT 2: After defining physical fitness, the definition of physical activity is mentioned, so content on the relationship between the two should be added.
RESPONSE 2: Thank you for your pertinent comment, we have added some information about this relationship in the introduction section.
COMMENT 3: Before the fact that there was a decrease in PA during COVID-19, the basis for problems that may arise due to the decrease in PA should be presented.
RESPONSE 3: Thank you for pointing this out. We have added information about this in the introduction.
COMMENT 4: Present a specific research hypothesis or research question.
RESPONSE 4: Thank you for your comments. We have taken care to better clarify our research objective in the introduction.
COMMENT 5: Can you provide information on how well the participants learned the measurement method through training? Since the degree of familiarity with the measurement method through training can affect the research results, it is thought that information on this should be provided.
RESPONSE 5: You are indeed right, we had not thought about assessing the degree of learning and familiarity with the test administration, but your observation is definitely something we need to take into account for future experiments.
COMMENT 6: More detailed descriptions should be given on how randomization was done.
RESPONSE 6: The same software was used for randomisation as for statistical analysis (SPSS = Statistical Package for Social Science). We have included this information in the protocol section.
COMMENT 7: As with the introduction, the results and discussion should be divided into paragraphs centered around the main results.
RESPONSE 7: Thank you for the advice. We have made the changes to the text that are in line with your wishes for a more readable text.
COMMENT 8: Can you see the difference in results by gender? Can you see the difference by bachelor and master? The above information is thought to be helpful in suggesting specific measures.
RESPONSE 8: We appreciate your comments, but in our specific case, our main goal was not to evaluate overall performance, but rather to investigate the consistency of the test. Accordingly, it would be useful to investigate whether discrepancies in error and accuracy between men and women fluctuate according to experimental conditions.
COMMENT 9: A note should be provided on the abbreviations used in the table.
RESPONSE 9: Thank you. We have added a note in the caption of each table to make them easier to read.
COMMENT 10: The detailed discussion of results should be omitted from the discussion.
RESPONSE 10: Thank you for your valuable feedback. We have carefully revised the discussion to streamline the content, removing any unnecessary details. By incorporating the changes you suggested, we were able to better focus and emphasize the central goal of our work, providing greater clarity and consistency.
COMMENT 11: There is no section on limitations of the study and suggestions for investing in platforms and technologies for remote test administration
RESPONSE 11: Thank you again for your suggestion. Now, in the text, there are both sections.

Round 2
Reviewer 1 Report
Comments and Suggestions for Authors
Authors have answered all of my comments accordingly.
I recommend that this paper can be publish in the current form.
Minor editing of English language required.
Reviewer 2 Report
Comments and Suggestions for Authors
Dear authors, after revising your paper, I consider it for publication.
Please, in case this paper to be published, you need to carefully improve the English quality of the text, maybe consulting an English professional to help you.
No further comments.
Comments on the Quality of English LanguageEnglish needs major improvements in case if the manuscript is accepted for publication.
Reviewer 3 Report
Comments and Suggestions for Authors
The comments were generally well reflected. It is regrettable that the results analysis according to the characteristics of the research participants was not presented.